# Desiccation resistance differences in *Drosophila* species can be largely explained by variations in cuticular hydrocarbons

**Zinan Wang[1,2]\*, Joseph P Receveur[1,2,3], Jian Pu[1,4], Haosu Cong[1], Cole Richards[1], Muxuan Liang[5], Henry Chung[1,2]\***

[1]Department of Entomology, Michigan State University, East Lansing, United States; [2]Ecology, Evolution, and Behavior Program, Michigan State University, East Lansing, United States; [3]Institute for Genome Sciences, University of Maryland, Baltimore, United States; [4]College of Agriculture, Sichuan Agricultural University, Sichuan, China; [5]Department of Biostatistics, University of Florida, Gainesville, United States

**\*For correspondence:**
wangzina@msu.edu (ZW);
hwchung@msu.edu (HC)

**Competing interest:** The authors declare that no competing interests exist.

**Abstract** Maintaining water balance is a universal challenge for organisms living in terrestrial environments, especially for insects, which have essential roles in our ecosystem. Although the high surface area to volume ratio in insects makes them vulnerable to water loss, insects have evolved different levels of desiccation resistance to adapt to diverse environments. To withstand desiccation, insects use a lipid layer called cuticular hydrocarbons (CHCs) to reduce water evaporation from the body surface. It has long been hypothesized that the water-proofing capability of this CHC layer, which can confer different levels of desiccation resistance, depends on its chemical composition. However, it is unknown which CHC components are important contributors to desiccation resistance and how these components can determine differences in desiccation resistance. In this study, we used machine-learning algorithms, correlation analyses, and synthetic CHCs to investigate how different CHC components affect desiccation resistance in 50 *Drosophila* and related species. We showed that desiccation resistance differences across these species can be largely explained by variation in CHC composition. In particular, length variation in a subset of CHCs, the methyl-branched CHCs (mbCHCs), is a key determinant of desiccation resistance. There is also a significant correlation between the evolution of longer mbCHCs and higher desiccation resistance in these species. Given that CHCs are almost ubiquitous in insects, we suggest that evolutionary changes in insect CHC components can be a general mechanism for the evolution of desiccation resistance and adaptation to diverse and changing environments.

## Editor's evaluation

These studies have presented convincing evidence that desiccation resistance in *Drosophila* species is conferred primarily by long, methyl-branched cuticular hydrocarbons. These fundamental findings add significantly to our understanding of how *Drosophila* species have evolved to adapt and survive in different environments. This study will be of interest to insect physiologists and ecologists as well as evolutionary biologists.

## Introduction

Maintaining water balance is a universal challenge for organisms living in terrestrial environments where water constantly evaporates from the body surface (*Hadley, 1994*). Organisms such as insects have a high surface area to volume ratio due to their small sizes, rendering them vulnerable to water loss and desiccation (*Gibbs, 2002b*; *Gibbs and Rajpurohit, 2010*; *Kühsel et al., 2017*). Studies using closely related insect species collected worldwide showed that phylogenetically related insect species can have very different levels of desiccation resistance and occupy very distinct habitats, while insects that are phylogenetically distant but dwelling in similar habitats could have similar levels of desiccation resistance (*Kellermann et al., 2012*; *Li et al., 2022*; *Menzel et al., 2017*; *Rane et al., 2019*). This suggests that extant species have evolved different levels of desiccation resistance to survive in their different habitats. However, as climate change accelerates the expansion of dryland (*Huang et al., 2016*) and changes aridity in many areas across the globe (*Sherwood and Fu, 2014*; *Shi et al., 2021*), it is less understood how insect species, which are integral parts of our ecosystems, can evolve higher levels of desiccation resistance to adapt to the more arid environments.

To determine how insects can adapt to desiccation stress, understanding how insects conserve water and prevent desiccation is crucial. In insects, cuticular water loss is the leading cause of desiccation (*Chown et al., 2011*; *Gibbs and Matzkin, 2001*; *Wang et al., 2021*). In an extreme example, cuticular water transpiration has been found to account for 97% of increased water loss in queens of the harvester ant *Pogonomyrmex barbatus* (*Johnson and Gibbs, 2004*). To conserve water and prevent desiccation, a general mechanism in insects is the use of a lipid layer on the epicuticle, named cuticular hydrocarbons (CHCs) (*Gibbs et al., 2003*; *Gibbs and Matzkin, 2001*). This CHC layer, which can contain more than 100 different compounds on the same individual, provides a hydrophobic barrier against evaporative water loss through the cuticle (*Blomquist and Ginzel, 2021*). This was first demonstrated when the physical or chemical removal of this layer using abrasive dust and various solvents resulted in increased water loss in various insect species in experiments published almost eight decades ago (*Wigglesworth, 1945*). In recent years, after the identification of a cytochrome P450 decarbonylase responsible for the synthesis of insect CHCs (*Qiu et al., 2012*), genetic manipulations of this gene in the fruitfly *Drosophila melanogaster* as well as in aphids and cockroaches, led to almost complete loss of CHCs and significant decrease in desiccation resistance (*Chen et al., 2016*; *Chen et al., 2019*; *Qiu et al., 2012*).

Variations in the composition of this CHC layer have been suggested to contribute to intraspecific variation in desiccation resistance in *D. melanogaster* (*Rouault et al., 2004*), the Mediterranean dung beetle *Onthophagus taurus* (*Leeson et al., 2020*), and the Argentine ant *Linepithema humile* (*Buellesbach et al., 2018*). This is supported by studies manipulating CHC composition either chemically or genetically, in different *Drosophila* species, producing different levels of desiccation resistance (*Chiang et al., 2016*; *Ferveur et al., 2018*; *Chung et al., 2014*; *Koto et al., 2019*; *Savage et al., 2021*). Together, these studies suggest that different components in the CHC layer can influence desiccation resistance.

The varying ability of these CHC components to prevent desiccation depends on its chemical structure, which in turn determines its melting temperatures (*Gibbs and Pomonis, 1995*; *Gibbs, 1998*; *Gibbs, 2002a*). The melting temperature of the hydrocarbon is positively correlated with its water-proofing properties and contribution to desiccation resistance (*Blomquist and Ginzel, 2021*; *Gibbs and Rajpurohit, 2010*). In *Drosophila* and most other insects, CHCs range in lengths between approximately 21 and 50 carbons, and consist of linear alkanes (*n*-alkanes), alkenes (monoenes and dienes), and methyl-branched alkanes (mbCHCs) (*Blomquist and Ginzel, 2021*). Among them, *n*-alkanes have the highest melting temperature, followed by mbCHCs , monoenes, and dienes. Increase in CHC chain length can also increase melting temperature and potentially leads to higher desiccation resistance (*Gibbs and Pomonis, 1995*). This is consistent with laboratory selection experiments in *D. melanogaster* selecting for increased desiccation resistance resulted in longer carbon-chain CHCs in the desiccation selected flies than the control flies (*Gibbs et al., 1997*; *Kwan and Rundle, 2010*). Longer CHC chain length is also correlated with climatic factors such as higher temperature and lower precipitation (*Jezovit et al., 2017*; *Rouault et al., 2004*). These climatic factors are associated with desiccation resistance in *Drosophila* species (*Hoffmann, 2010*; *Hoffmann and Weeks, 2007*; *Kellermann et al., 2018*).

While the studies mentioned above showed strong evidence that CHCs and variations in CHC compositions are associated with desiccation resistance in insects, no study has investigated the extend that CHC variation can determine desiccation resistance and whether we can identify the important CHC components that underlie variation in desiccation resistance across different species. Several large-scale studies produced large datasets in determining the evolution of desiccation resistance across closely related *Drosophila* species (**Kellermann et al., 2018**; **Kellermann et al., 2012**; **Matzkin et al., 2009**) and ant species (**Bujan et al., 2016**; **Hood and Tschinkel, 1990**). Other studies produced large datasets in CHC variation across species (**Khallaf et al., 2021**; **Menzel et al., 2017**; **Nunes et al., 2017**; **Van Oystaeyen et al., 2014**) and focused on the communication aspects of CHCs. However, variations in CHCs and differences in desiccation resistance have not been experimentally connected in a phylogenetic framework. Analyzing CHC profiles and desiccation resistance across closely related species measured under the similar conditions can determine important CHC components that could predict various levels of desiccation resistance. Understanding the mechanisms underlying desiccation resistance and examining the evolution of these traits is key to predicting species' responses in facing drier environments that could result from future climate change (**Chown et al., 2010**).

In this study, we used a cohort-based experimental design of 50 *Drosophila* and related species, and experimentally determined their CHC compositions and desiccation resistance under similar conditions. Using a random forest machine-learning algorithm, we built decision trees connecting CHC components and desiccation resistance in the 50 species and tested their correlation. First, we determined that CHC variations can largely explain differences in desiccation resistance across these species. Second, we identified a subset of CHCs, the methyl-branched CHCs (mbCHCs), as the most important CHC components that can determine desiccation resistance across these species. Third, we determined that the evolution of longer mbCHCs is also significantly correlated with evolution of higher desiccation resistance in *Drosophila* species. Taken together, our study showed that mbCHCs are a key predictive component of desiccation resistance in *Drosophila* species and suggests that evolution in CHC components can be a general mechanism for higher desiccation resistance and adaptation to more arid environments.

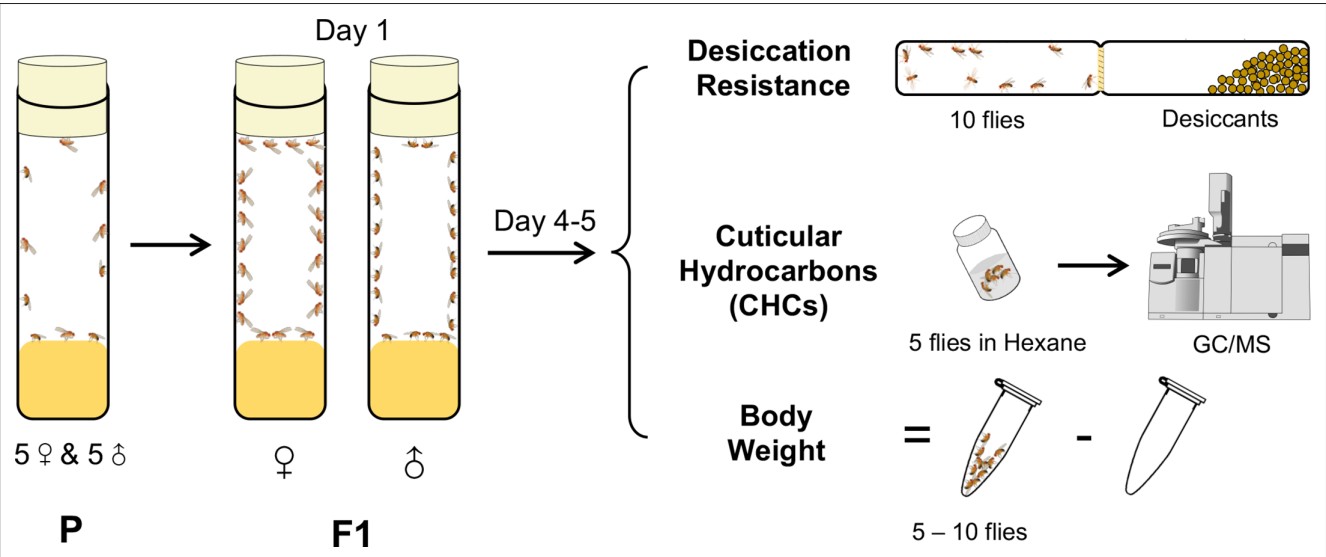

**Figure 1.** A cohort-based experimental design. A cohort-based experimental design can facilitate determining the correlation between desiccation resistance, cuticular hydrocarbons (CHCs), and body weight. Five to six cohorts of each species were established.

The online version of this article includes the following figure supplement(s) for figure 1:

**Figure supplement 1.** The phylogeny of 50 *Drosophila* and related species used in this study.

**Figure supplement 2.** Chromatogram of the authentic standard *n*-alkane mixture which contains 34 *n*-alkanes from C7 to C40 at the same concentration.

## Results

### A cohort-based experimental design for determining correlations between CHCs and desiccation resistance

To investigate how CHC variation across an evolutionary gradient affects desiccation resistance, for our experiments, we selected 46 *Drosophila* species representing both the *Sophophora* subgenus and the *Drosophila* subgenus, as well as three *Scaptodrosophila* species and one *Chymomyza* species (*Figure 1—figure supplement 1*). As the assays for CHCs and desiccation resistance cannot be performed on the same individuals, we used a cohort-based experimental design (*Figure 1*) where subsets of each cohort (5–6 per species) were used for gas chromatography–mass spectrometry (GC–MS) determination of CHC profiles, desiccation resistance assays, and measuring body weight of the F1 progeny for each sex. Therefore, all measurements were performed at the cohort level.

Desiccation resistance varied across these 50 species with desert-dwelling species from the repleta group showing the highest desiccation resistance (e.g., *D. mojavensis* males: 58.2 ± 4.7 hr) and species from *melanogaster* group showing the lowest desiccation resistance (e.g., *D. mauritiana* males: 2.7 ± 0.2 hr) (*Figure 2*, *Figure 2—figure supplement 1*), consistent with the findings of previous research (*Kellermann et al., 2012*; *Matzkin et al., 2009*). GC–MS analyses of the CHCs in these 50 species detected five types of CHCs with different carbon-chain lengths and quantities, including linear alkanes (*n*-alkanes), methyl-branched alkanes (mbCHCs; the methyl branch is on the second carbon), monoenes (with one double bond), dienes (with two double bonds), and trienes (with three double bonds) (*Figure 2—figure supplement 2*). *n*-Alkanes, which have the highest melting temperatures, were only present in species from the *melanogaster* group, some of which have the lowest desiccation resistance among the species tested in this study. This suggests that *n*-alkanes may not have a general contribution to desiccation resistance across *Drosophila* species. Trienes, which have the lowest melting temperatures, were only present in eight species in low-to-moderate quantities. The other three types of CHCs, the mbCHCs, monoenes, and dienes, were observed in most species tested in our study.

### Higher relative quantities of CHCs are not primarily responsible for higher desiccation resistance

We first sought to determine if higher desiccation resistance in flies could be due to having higher amounts of CHCs. One caveat in the analysis between absolute CHC quantity and desiccation resistance is a possible correlation between CHC quantities and body size: species with larger body size may possess higher quantities of CHCs. This can lead to potential biases when directly determining correlations between CHC quantities and desiccation resistance. Species with higher body weight can also have higher absolute water content and a lower surface area to volume ratio due to their larger size, which may also lead to higher desiccation resistance (*Wang et al., 2021*).

Our initial analyses showed a significant positive correlation (Females: $r = 0.4$, $p < 0.001$; Males: $r = 0.5$, $p < 0.001$) between body weight and desiccation (*Figure 3A*). We also found a significant positive correlation between body weight and total amount of CHCs (Female: $r = 0.7$, $p < 0.001$; Male: $r = 0.7$, $p < 0.001$) (*Figure 3B*), suggesting that variation in the body weight (or size) of different species may be a confound in determining the relationship between CHC quantity and desiccation resistance. To take this confound into consideration, we normalized the total CHC quantity by dividing by the body weight for each species. This shows that CHCs account for 0.02–0.5% of the total body weight of different species (*Figure 3—figure supplement 1*). Analyses between the normalized CHC quantity and desiccation resistance showed no correlation between these two variables in males (Male: $p = 0.1$) and positive correlation in females (Female: $r = 0.1$, $p = 0.03$) (*Figure 3C*). The low value of $r$ (0.1) indicates a weak correlation between CHC quantities and desiccation resistance in females. This suggests that having higher amounts of CHCs only have a limited contribution to higher desiccation resistance.

### mbCHCs are important determinants of desiccation resistance

As total CHC quantity is not a main contributor to desiccation resistance in our study, we hypothesized that the composition of CHC profiles may be important for desiccation resistance. The CHC layer of each *Drosophila* species is composed of different CHC components, each with different levels of abundance, so we used 'beta diversity' to represent this variation in the CHC composition of our

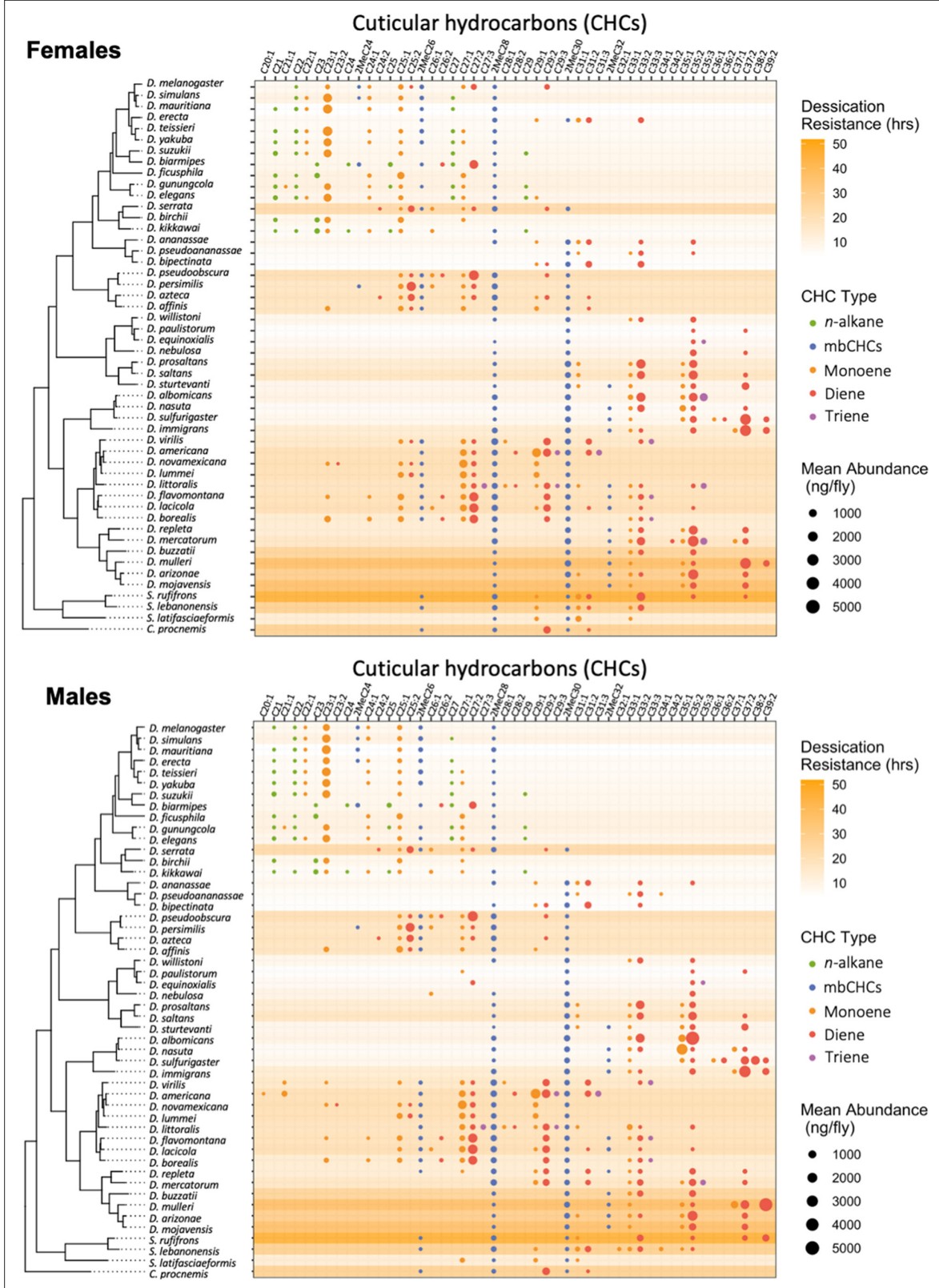

**Figure 2.** Desiccation resistance and cuticular hydrocarbon (CHC) composition in 46 *Drosophila* species and 4 outgroup species. Desiccation resistance and CHC composition of each species were plotted together. Males and females are shown separately. The shading intensity of each species represents the number of hours of desiccation resistance, while the size and color of each circle represent the type of CHC and its quantity. *n*-Alkanes are only present in some species from the *melanogaster* group.

*Figure 2 continued on next page*

*Figure 2 continued*

The online version of this article includes the following figure supplement(s) for figure 2:

**Figure supplement 1.** Desiccation resistance and body weight in 46 *Drosophila* species and 4 outgroup species.

**Figure supplement 2.** The melting temperature of cuticular hydrocarbons (CHCs) is determined by methyl group, double bonds, and carbon-chain length.

sampled species, similar to what is used in microbiome analysis. This allows us to test the correlation between the compositional variation and desiccation resistance for the sampled *Drosophila* species. We then applied a permutational multivariate analysis of variance (PERMANOVA) to test whether the composition of CHCs differed across desiccation resistance (*Anderson et al., 2006*). Our results showed that the beta diversity of CHC compositions differs significantly across the increasing desiccation resistance ($r^2$ = 0.1, p < 0.001; *Figure 4—figure supplement 1*), suggesting that CHC composition is important for desiccation resistance. We further sought to identify individual components of CHC profiles that are important determinants of desiccation resistance. To test if variation in CHC components can be used to determine desiccation resistance, we applied a random forest regression method that uses decision trees to connect the variables of interest (*Liaw and Wiener, 2002*; *Svetnik et al., 2003*), for example, CHC composition and desiccation resistance, and identified the importance of individual CHC components to predict desiccation resistance. In this analysis, we treated the CHC profiles of each species and sex as an individual dataset, giving us 100 datasets (50 species × 2 sexes) with five to six individual CHC profiles each. We then correlated the decision trees that generated from the CHC composition with desiccation resistance. The prediction process can identify key CHC components that are important contributors to desiccation resistance.

Random forest modeling of CHC composition was able to explain 85.5% of the variation in desiccation resistance (out of bag estimate: root mean square error 'RMSE' = 4.5) (*Figure 4A*). Models built with a 70:30 training/test split (n = 382, n = 164) performed similar to models built with out of bag estimate (RMSE = 5.4, *Figure 4—figure supplement 2*). Four mbCHCs were identified as having the highest contribution to predicting desiccation resistance in the regression model (listed in decreasing importance: 2MeC30, 2MeC28, 2MeC32, and 2MeC26), while many other CHCs did not substantially contribute to predicting desiccation resistance (*Figure 4B*). These results suggest that CHCs, and mbCHCs in particular, are important predictors of desiccation resistance in *Drosophila* species.

## Longer mbCHCs are correlated with higher desiccation resistance

We further sought to determine how mbCHCs contribute to desiccation resistance. As each species produces different combinations of these mbCHCs that are correlated with each other (*Figure 5—figure supplement 1*), we did not perform correlation analyses on these mbCHCs with a single regression model due to multicollinearity issues (*Belsley et al., 2005*). Instead, we performed correlation analyses between each of the most important CHCs identified from the random forest modeling, 2MeC26, 2MeC28, 2MeC30, and 2MeC32, and desiccation resistance. We found that the quantity of 2MeC26 negatively correlates with desiccation resistance in males (r = −0.4, p < 0.001) but no significant correlation was identified in females (p = 0.1) (*Figure 5A*). Significant positive correlations were observed between quantities of the longer mbCHCs and desiccation resistance for females (2MeC28: r = 0.4, p < 0.001; 2MeC30: r = 0.2, p = 0.009; 2MeC32: r = 0.3, p < 0.001) and males (2MeC28: r = 0.4, p < 0.001; 2MeC32: r = 0.4, p < 0.001) (*Figure 5B–D*). However, no significant correlation was observed between quantity of 2MeC30 and desiccation resistance in males (*Figure 5C*). These results suggested the production of longer mbCHCs could play a role in higher desiccation resistance.

## Longer saturated CHCs can confer higher desiccation resistance in wild-type *D. melanogaster*

To investigate the above results suggesting that longer mbCHCs could underlie higher desiccation resistance in *Drosophila* species, we performed desiccation assays on *D. melanogaster* (*attP40* strain) coated with synthetic mbCHCs of different lengths (2MeC26, 2MeC28, and 2MeC30). The *D. melanogaster attP40* strain is a common lab strain used for transgenesis with a wild-type CHC profile. While previous studies showed that a mixture of 2MeC26, 2MeC28, and 2MeC30 can slightly increase desiccation resistance in *D. melanogaster* flies without CHCs, experiments were not performed on

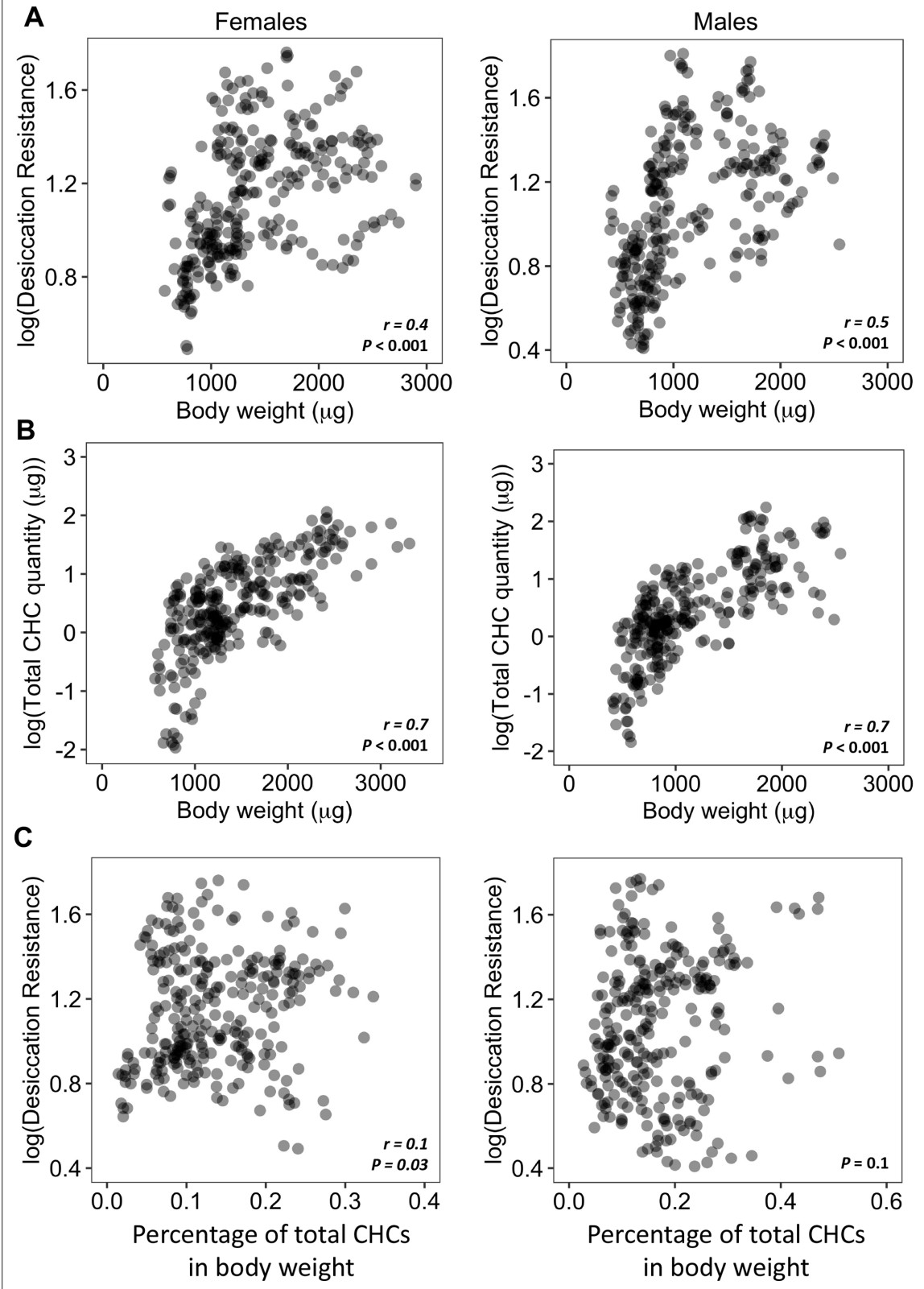

**Figure 3.** Higher amounts of cuticular hydrocarbons (CHCs) do not contribute to higher desiccation resistance. (**A**) Body weight is positively correlated with desiccation resistance (Females: $r = 0.4$, $p < 0.001$, Males: $r = 0.5$, $p < 0.001$). (**B**) Total amount of CHCs is correlated with having higher body weight (Females: $r = 0.7$, $p < 0.001$, Males: $r = 0.7$, $p < 0.001$). (**C**) A weak positive correlation between desiccation resistance and CHCs as a percentage of body

*Figure 3 continued on next page*

*Figure 3 continued*

weight in females, while no correlation in males (Females: *r* = 0.1, p = 0.03, Males: p = 0.1). All correlation analyses were conducted using Pearson's method.

The online version of this article includes the following figure supplement(s) for figure 3:

**Figure supplement 1.** Cuticular hydrocarbon (CHC) quantity as a percentage of body weight.

individual mbCHCs for desiccation resistance (*Krupp et al., 2020*). Our desiccation assays showed that the coating of 2MeC26 did not increase desiccation resistance in both sexes (post hoc comparison with Tukey's method followed by one-way analysis of variance (ANOVA). Female: p = 0.07; Male: p = 1.0), while 2MeC30 significantly increased desiccation resistance in female *D. melanogaster* (*t* = 11.2, p < 0.001), and both 2MeC28 (*t* = 4.6, p < 0.001) and 2MeC30 (*t* = 6.9, p < 0.001) significantly increased desiccation resistance in male *D. melanogaster* (*Figure 5E*). There is a positive correlation with longer mbCHC length coated and higher desiccation resistance (*Figure 5—figure supplement 2*), suggesting that having higher quantities of longer mbCHCs can lead to higher desiccation resistance. As a control, we repeated the same experiment with synthetic *n*-alkanes (C23, C25, C27, C29, and C31). Coating with *n*-alkanes on the *attP40* strain led to increases in desiccation resistance, which is positively correlated with carbon-chain length, similar to the mbCHCs experiments, suggesting that longer chain CHCs, with higher melting temperatures, can confer higher desiccation resistance when coated on the *attP40* strain (*Figure 5—figure supplement 3*, *Figure 5—figure supplement 4*).

The coating of synthetic mbCHCs and *n*-alkanes on the *attP40* strain showed that adding more CHCs to a strain with a wild-type CHC profile can increase desiccation resistance but did not show if these individual CHCs are sufficient to confer higher desiccation resistance. When these coating experiments (three mbCHCs, five *n*-alkanes) were individually repeated on *D. melanogaster* flies without CHCs (CHC-), we observed that while all mbCHCs and *n*-alkanes are able to confer higher desiccation resistance when coated on female CHC- flies, only a subset of these CHCs are able to confer resistance on male CHC- flies (*Figure 5—figure supplement 5*). In addition, only a positive correlation between longer mbCHCs and higher desiccation was observed in female CHC- flies. There is no correlation between mbCHC length and higher desiccation in male CHC- flies and no correlation between *n*-alkane length and higher desiccation resistance in female and male CHC- flies (*Figure 5—figure supplement 6*, *Figure 5—figure supplement 7*). We noted that coating mbCHCs and *n*-alkanes only produced a small increase in desiccation resistance in CHC- flies, compared to coating these CHCs on the *attP40* strain. This is similar to a previous study by *Krupp et al., 2020* where the authors showed similar small increases in desiccation resistance when only *n*-alkanes or mbCHCs are coated on CHC- flies, compared to the larger increase in desiccation resistance when wild-type extracts of CHCs were coated. Taken together, these results suggest that CHCs with higher melting temperatures such as longer mbCHCs or *n*-alkanes contribute to higher desiccation resistance when in a blend of CHCs but not sufficient to confer high desiccation resistance by themselves. This is consistent with the hypotheses from multiple papers that the cuticular lipid layer works as a blend or a mixture rather than as individual components (*Gibbs, 1998*; *Menzel et al., 2019*; *Wigglesworth, 1945*). Although we showed that both longer mbCHCs and *n*-alkanes can confer higher desiccation resistance when coated on *D. melanogaster* with a wild-type profile, as *n*-alkanes are mostly absent in the *Drosophila* genus, our experiments suggest a causal relationship between longer mbCHCs and higher desiccation resistance in *Drosophila*.

## Evolution of mbCHCs underlies variation in desiccation resistance in *Drosophila* species

We have shown that mbCHCs are important determinants of desiccation resistance in *Drosophila* species and experimentally coating insects with longer mbCHCs confers higher desiccation resistance. To further determine whether the evolution of mbCHCs underlies the variation in desiccation resistance in *Drosophila* species, we first examined the evolutionary trajectory of mbCHCs and then tested the correlation between mbCHCs and desiccation resistance with phylogenetic correction. We mapped the evolution of mbCHC composition in the *Drosophila* genus using ancestral trait reconstruction and tested the phylogenetic signal using Pagel's $\lambda$ (*Freckleton et al., 2002*; *Goolsby et al., 2017*). Ranging from 0 to 1, Pagel's $\lambda$ measures the extent to which the variance of the trait can be explained

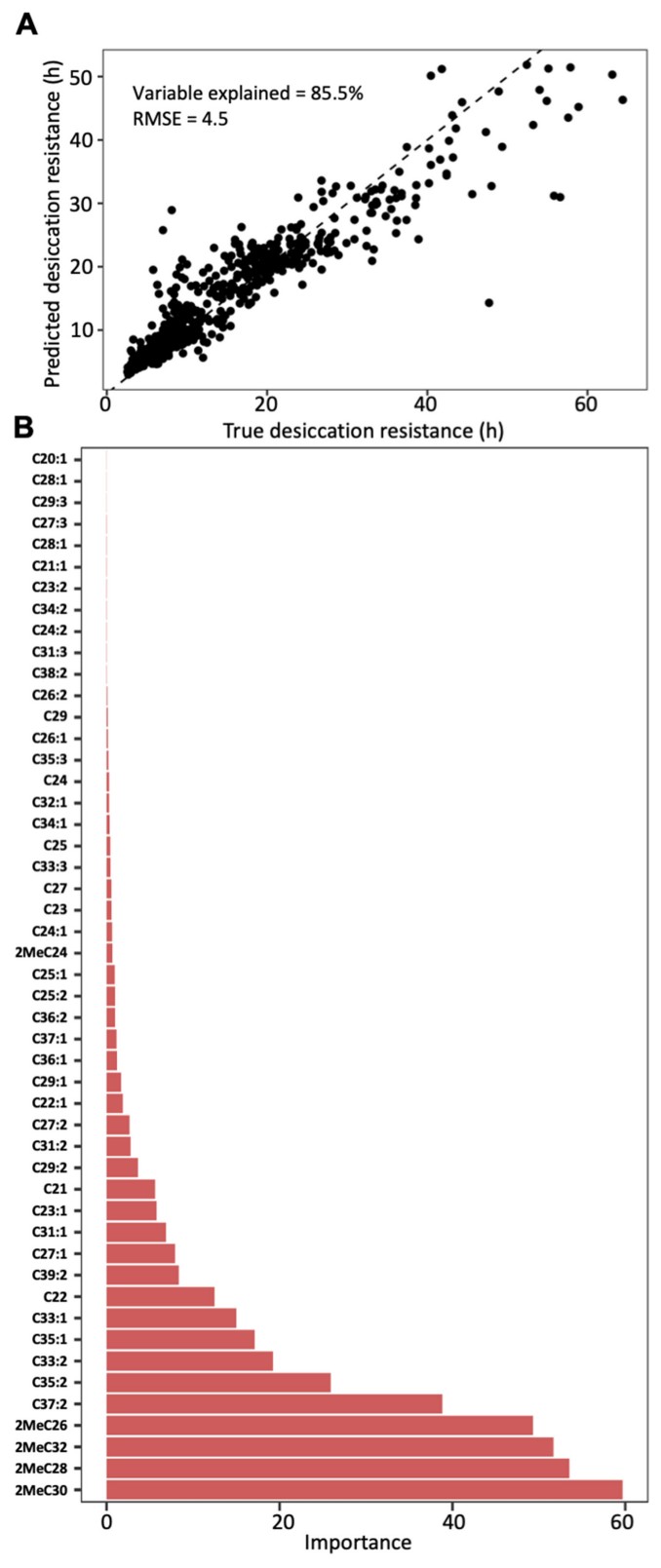

**Figure 4.** Cuticular hydrocarbon (CHC) composition can be used to predict desiccation resistance. (**A**) Random Forest regression modeling of CHC abundance was able to explain 85.5% of the variation in time to desiccation with a root mean square error (RMSE) of 4.5. (**B**) The abundance of four mbCHCs, 2MeC30, 2MeC28, 2MeC32, and

*Figure 4 continued on next page*

*Figure 4 continued*

2MeC26, has the highest importance to the desiccation resistance in the random forest regression model, while most of CHCs have less contribution to the accuracy of the model for desiccation resistance.

The online version of this article includes the following figure supplement(s) for figure 4:

**Figure supplement 1.** The composition of cuticular hydrocarbons (CHCs) significantly differed across the increasing desiccation resistance.

**Figure supplement 2.** Cross-validation of the random forest regression model has a similar performance to the regression model using the full dataset.

by the phylogeny and therefore determines the degree of association between the trait evolution and the phylogeny (*Lynch, 1991*). If $\lambda$ is closer to 1, the evolution of the trait has a higher association with the phylogeny. Phylogenetic signals in mbCHCs were detected for both females and males with $\lambda = 0.75$ and 0.83, respectively (*Supplementary file 1a*). This suggests a moderate to strong association between mbCHC evolution and phylogeny. The derived ancestral state for mbCHCs in the *Drosophila* genus suggest that 2MeC28 and 2MeC30 are the major mbCHCs (*Figure 6—figure supplement 1*). The evolutionary trajectory of mbCHC composition showed repeated evolution in the lengths of mbCHCs, as well as their quantities. Independent evolution of longer mbCHCs was observed in four clades of species, including the willistoni, nasuta, and repleta groups, and the ananassae subgroup (including *D. ananassae*, *D. pseudoananassae*, and *D. bipectinata*), while shorter mbCHCs were mainly observed in several species in the *melanogaster* subgroup (e.g., *D. melanogaster*, *D. simulans*, and *D. teissieri*) (*Figure 2*, *Figure 6—figure supplement 1*).

Desiccation resistance also has a strong association with the phylogeny of *Drosophila* species (*Supplementary file 1b*; *Kellermann et al., 2018*; *Kellermann et al., 2012*). We sought to determine if the variation of desiccation resistance could be explained by the evolution of mbCHCs in these species. We used a Phylogenetic Generalized Linear Square (PGLS) model to test for the correlation between mbCHCs and desiccation resistance when controlling the effects from the phylogenetic relationship of the species (*Grafen, 1989*; *Mundry, 2014*). Since mbCHCs are produced in a linear pathway with each species producing several different mbCHCs, we selected the mbCHC with the longest carbon-chain length in the species as the proxy to represent the lengths of mbCHCs in each species. We incorporated the quantity, length, and their interaction in the PGLS model for the longest mbCHC. The interaction term between the length and quantity of the longest mbCHC in the model can determine how the two variables combinatorically affect desiccation resistance. PGLS modeling showed, after correcting for phylogenetic effects, the higher quantity and longer length of the longest mbCHC both affect desiccation resistance (interaction term, Female: $t = 3.5$, p < 0.001; Male: $t = 2.2$, p = 0.03) (*Supplementary file 1c*; *Figure 6*). This suggests that the synthesis of mbCHCs with longer carbon-chain lengths could be a common mechanism underlying the evolution of higher desiccation resistance.

## Discussion

Reducing evaporative water loss through the cuticle using a layer of CHCs is one of the most important evolutionary innovations in insects that allows many species to survive and thrive in diverse and arid habitats. We showed that CHC composition can account for 85.5% of the variation in desiccation resistance in the 50 *Drosophila* and related species in this study. This suggests that CHC composition may be highly predictive of desiccation resistance. Algorithmic ranking in importance of CHC components using a random forest machine-learning model showed that mbCHCs have the highest contribution to determining desiccation resistance. Importantly, higher amounts of longer mbCHCs are important in desiccation resistance. This is consistent with previous studies showing that mbCHCs have higher melting temperatures than the other commonly present types of CHCs (monoenes and dienes) and longer CHC leads to higher melting temperature (*Gibbs and Pomonis, 1995*). Our overall analyses support the conclusion that the evolution of longer mbCHCs is correlated with higher desiccation resistance in our dataset, but we observed minor differences between males and females (*Figures 3 and 5*). In particular, while higher quantities of 2MeC28 and 2MeC32 are both positively correlated with higher desiccation resistance in both males and females (*Figure 5B, D*), there are differences between the sexes regarding 2MeC26 and 2MeC30 (*Figure 5A, C*). There could be several underlying

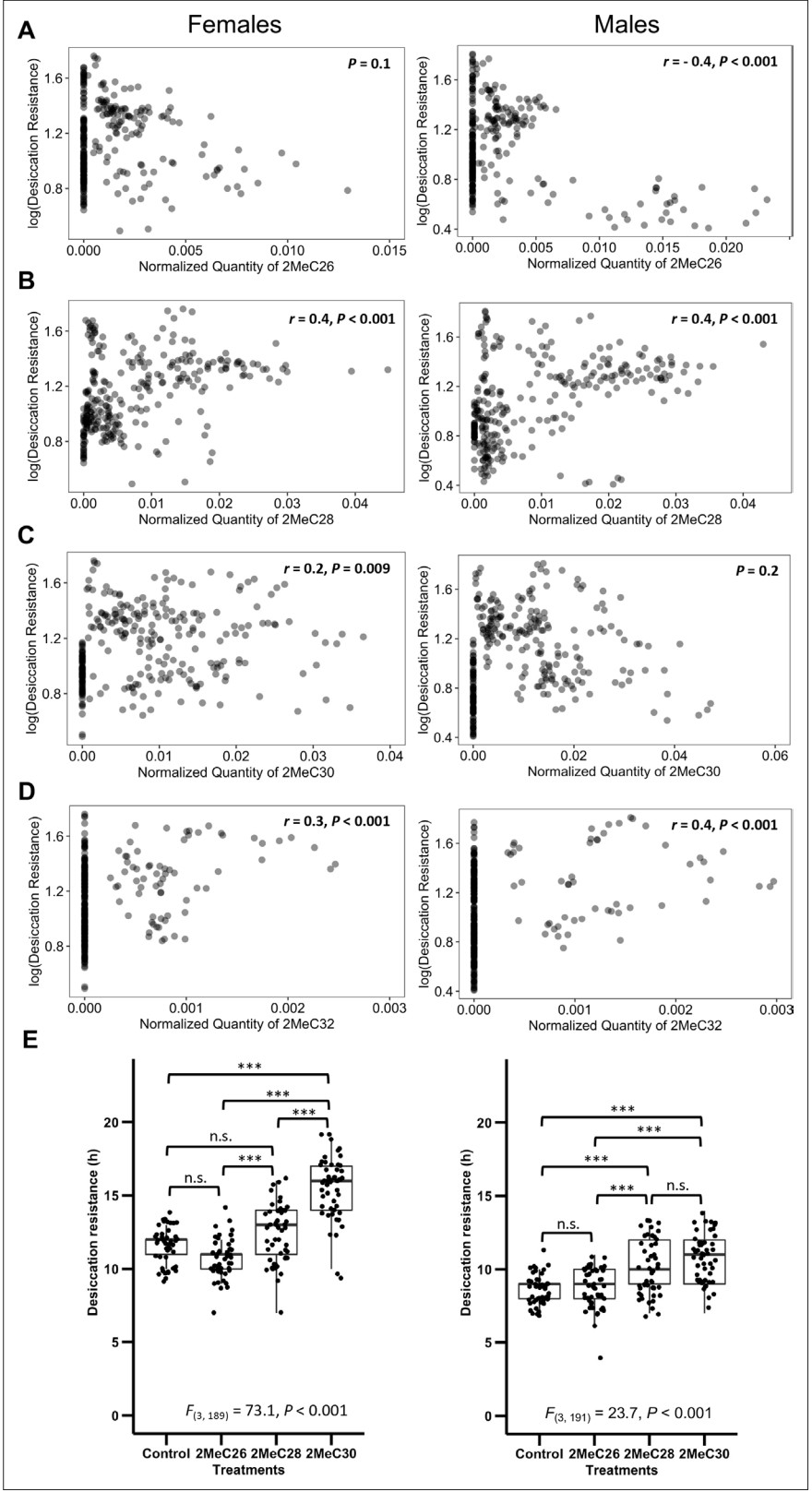

**Figure 5.** Longer mbCHCs are associated with higher desiccation resistance. (**A**) Quantity of 2MeC26 is negatively correlated with desiccation resistance in males, but no correlation in females (Females: p = 0.1, Males: $r = -0.4$, p < 0.001). (**B**) Quantity of 2MeC28 is positively correlated with desiccation resistance (Females: $r = 0.4$, p < 0.001, Males: $r = 0.4$, p < 0.001). (**C**) Quantity of 2MeC30 is positively correlated with desiccation resistance in females,

*Figure 5 continued on next page*

*Figure 5 continued*

but no correlation in males (Females: $r = 0.2$, $p = 0.009$, Males: $p = 0.2$). (**D**) Quantity of 2MeC32 is positively correlated with desiccation resistance (Females: $r = 0.3$, $p < 0.001$, Males: $r = 0.4$, $p < 0.001$). Correlations between the quantities of each mbCHC and desiccation resistance were determined using Pearson's method. (**E**) *D. melanogaster* coated with longer mbCHCs have higher desiccation resistance. Coating of synthetic mbCHCs on *D. melanogaster* showed that 2MeC26 does not influence desiccation resistance. 2MeC30 increases desiccation resistance in female *D. melanogaster* while both 2MeC28 and 2MeC30 in increases desiccation resistance in *male D. melanogaster*. The bold horizontal line within each box plot corresponds to the median value, the box length to the interquartile range, and the lines emanating from the box (whiskers) extend to the smallest and largest observations. One-way analysis of variance (ANOVA) showed significant differences between *D. melanogaster* flies coated with different mbCHCs (Female: $F_{(3,189)} = 73.1$, $p < 0.001$, Male: $F_{(3,191)} = 23.7$, $p < 0.001$). Post hoc comparison was conducted using Tukey's method. ***$p < 0.001$; n.s. refers to not significant.

The online version of this article includes the following source data and figure supplement(s) for figure 5:

**Source data 1.** Desiccation resistance of *Drosophila melanogaster attP40* flies when coated with individual mbCHC of 2MeC26, 2MeC28, and 2MeC30.

**Source data 2.** Desiccation resistance of *Drosophila melanogaster attP40* flies when coated with individual *n*-alkane of C23, C25, C27, C29, and C31.

**Source data 3.** Desiccation resistance of *Drosophila melanogaster* CHC- flies when coated with individual CHC of 2MeC26, 2MeC28, 2MeC30, C23, C25, C27, C29, and C31.

**Figure supplement 1.** Correlation between different mbCHCs in 50 *Drosophila* and related species.

**Figure supplement 2.** The length of coated mbCHCs on *D. melanogaster attP40* flies is positively correlated with desiccation resistance in both females and males.

**Figure supplement 3.** Desiccation resistance of *D. melanogaster attP40* flies coated with individual *n*-alkanes (C23, C25, C27, C29, and C31).

**Figure supplement 4.** The length of coated *n*-alkanes on *D. melanogaster attP40* flies is positively correlated with desiccation resistance in both females and males.

**Figure supplement 5.** Desiccation resistance of CHC- *D. melanogaster* flies coated with individual mbCHCs (2MeC26, 2MeC28, 2MeC30) or *n*-alkanes (C23, C25, C27, C29, and C31).

**Figure supplement 6.** The length of coated mbCHCs on CHC- flies has a weak positive correlation with desiccation resistance in females but did not correlate with the levels of desiccation resistance in males.

**Figure supplement 7.** The length of coated *n*-alkanes on CHC- flies did not correlate with desiccation resistance.

**Figure supplement 8.** Pathway for the synthesis of branched cuticular hydrocarbons (mbCHCs) and linear CHCs (*n*-alkanes, monoenes, and dienes).

**Figure supplement 9.** Gas chromatography–mass spectrometry (GC–MS) chromatograms of *5'mFAS*-GAL4, UAS-*Cyp4g1*-RNAi, and the F1 offspring from *5'mFAS*-GAL4 × UAS-*Cyp4g1*-RNAi.

reasons for this. Firstly, across our dataset, CHCs are sexually dimorphic in many species, leading to overall differences in the correlation analyses between the sexes. Secondly, other physiological differences, such as size differences, may contribute to the differences between males and females. In many species such as *D. melanogaster*, females are larger than males and this may contribute to the higher basal desiccation resistance (when CHCs are taken out of the equation) of female CHC- flies (3.6 ± 0.1 hr) compared to that of male CHC- flies (2.1 ± 0.1 hr) (*Figure 5E*). While these are minor differences in the overall correlation analyses between males and females, the synthetic CHC coating experiments showed that coating of longer mbCHCs confers higher desiccation resistance in both males and females.

## mbCHCs and desiccation resistance

Previous studies showed support for mbCHCs in desiccation resistance in *Drosophila*. RNAi knock-down of a methyl-branched specific fatty acid synthase (*mFAS*) in *D. serrata* eliminates almost all mbCHCs and leads to significant decrease in desiccation resistance which could be partially rescued with synthetic mbCHCs (*Chung et al., 2014*) . Its sibling species, the rainforest desiccation sensitive *D. birchii*, have only trace amounts of mbCHCs (*Howard et al., 2003*) and is unable to evolve desiccation resistance despite strong laboratory selection over many generations (*Hoffmann et al., 2003*). The presence of mbCHCs in most *Drosophila* and related species leads to another question:

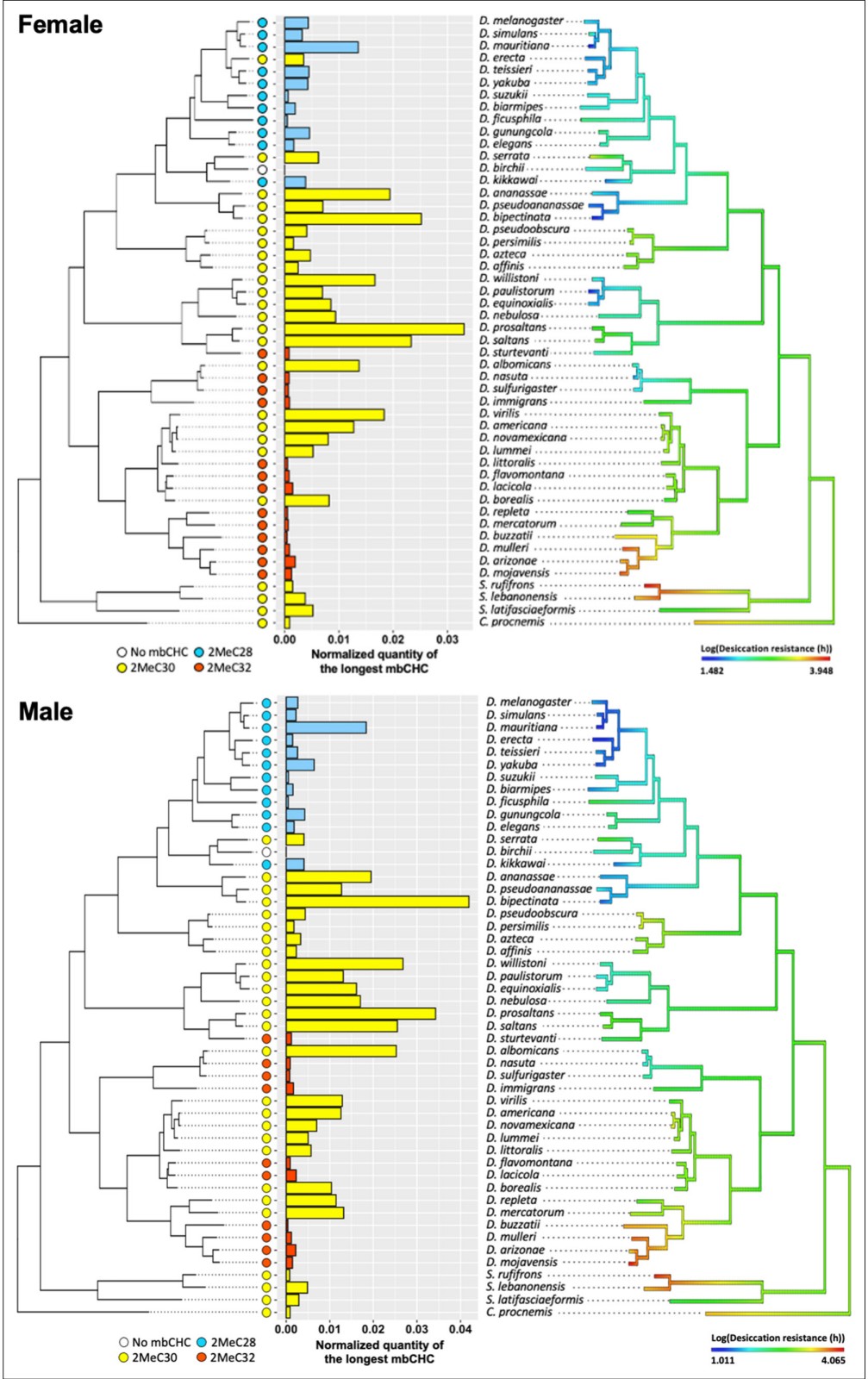

**Figure 6.** Evolution of longer mbCHCs is significantly correlated with evolution of higher desiccation resistance. Patterns in the normalized quantities of the longest mbCHCs and desiccation resistance for females (top) and males (bottom) were listed across the phylogeny of the 50 *Drosophila* and related species. Phylogenetic Generalized Linear Square (PGLS) analysis between the longest mbCHCs and desiccation resistance showed

*Figure 6 continued on next page*

*Figure 6 continued*

both the higher quantity and longer length of the longest mbCHC affect desiccation resistance (interaction term, Female: $t = 3.5$, $p < 0.001$; Male: $t = 2.2$, $p = 0.03$).

The online version of this article includes the following source data and figure supplement(s) for figure 6:

**Source data 1.** Normalized quantities of mbCHCs, the length and normalized quantities of the longest mbCHC in each species.

**Figure supplement 1.** Ancestral state reconstruction of mbCHC for the *Drosophila* genus.

Is this a general rule for insects using mbCHCs to develop desiccation resistance? A high proportion of very-long-chain mbCHCs with a single methyl branch have been reported in many insect species from different taxa that dwell in desert environments, such as the desert tenebrionid beetle *Eleodes armata* (**Hadley, 1977**), the desert ants *Cataglyphis niger* and *P. barbatus* (**Johnson and Gibbs, 2004**; **Soroker and Hefetz, 2000**), and the desert locust *Schistocerca gregaria* (**Heifetz et al., 1998**). Although a small proportion of *n*-alkanes was also reported in some of these species, this suggests the use of mbCHCs to minimize water evaporation may be a common mechanism for insects in extremely dry environments.

Ancestral trait reconstruction showed that the derived ancestral status for mbCHCs in the *Drosophila* genus is 2MeC28 and 2MeC30 as the major mbCHCs (**Figure 6—figure supplement 1**). As there are many extant species that do not have 2MeC30, this suggests that the ability to synthesize longer mbCHCs may be lost during evolution, especially for species that do not inhabit arid environments and have low levels of desiccation resistance such as *D. melanogaster* and *D. biarmipes*. On the contrary, for species that dwell in extremely arid environments such as *D. mojavensis* and *D. arizonae*, higher quantities of 2MeC30 and an even longer mbCHC, 2MeC32, are observed. The observation is consistent with the results from our PGLS modeling, showing significant correlation between having higher quantity of longer mbCHCs and higher desiccation resistance in the *Drosophila* genus.

## Constraints in evolution of CHCs and desiccation resistance

Why are *n*-alkanes, which have the highest melting temperature and potentially better water-proofing properties, not prevalent in species in our studies? A hypothesis that could explain this is the competition on the common precursors between the synthesis of linear unsaturated CHCs (monoenes and dienes) and *n*-alkanes. In insects, the synthesis of all CHCs start from the fatty acyl CoA synthesis pathway. The pathway is split where a cytosolic fatty acid synthase (cFAS) will synthesize all the linear CHCs (*n*-alkanes, monoenes, dienes), while mFAS will synthesize mbCHCs (**Figure 5—figure supplement 8**; **Chung and Carroll, 2015**; **Holze et al., 2021**). The pathway suggests that the synthesis of *n*-alkanes, monoenes, and dienes are in competition. As many monoenes and dienes function in *Drosophila* as contact pheromones in regulating different behaviors such as mating and aggression (**Blomquist and Ginzel, 2021**; **Chertemps et al., 2007**; **Chertemps et al., 2006**; **Chung and Carroll, 2015**; **Krupp et al., 2008**; **Wang et al., 2011**), the synthesis for these CHCs may compete with the synthesis of *n*-alkanes. In contrast, the synthesis of mbCHCs is in another part of the pathway. This suggests that there are potential constraints in the roles of *n*-alkanes in desiccation resistance in *Drosophila* due to competition with CHCs which function as signaling molecules, while the synthesis of mbCHCs is relatively unaffected by other products of the pathway (**Figure 5—figure supplement 8**; **Chung and Carroll, 2015**). We speculate that in *Drosophila* and related species, the use of mbCHCs for modulating desiccation resistance may avoid the conflict between surviving desiccation and chemical signaling.

If there are no biosynthetic constraints to the synthesis of mbCHCs, would the evolution of longer mbCHCs be a general mechanism in the evolution of higher desiccation for species adapting to more arid climates? We suggest that this may be the case for many species, but for some species that use the shorter mbCHCs such as 2MeC26 and 2MeC28 as signaling molecules, such as *D. serrata* (**Chenoweth and Blows, 2005**; **Chung et al., 2014**) and the longhorned beetle, *Mallodon dasystomus* (**Spikes et al., 2010**), natural selection and sexual selection may have opposing effects on the lengths of these mbCHCs. In *D. serrata*, having higher amounts of 2MeC26 leads to higher mating success in male *D. serrata* (**Chung et al., 2014**). *Northern* populations of this species produce lower amounts of 2MeC26 but higher amounts of 2MeC30, due to a polymorphism in a fatty acyl-CoA reductase

gene that shifts to an overall production of longer CHCs. As CHCs are sexually monomorphic in *D. serrata*, this polymorphism led to lower mating success in male *northern* flies and higher desiccation resistance in female *northern* flies compared to their common counterparts (*Rusuwa et al., 2022*). The sexually antagonistic effect due to the functions of the different mbCHCs may affect their evolutionary response to desiccation in many species.

## CHC variation and desiccation resistance across insects

How much of what we learnt in this study is applicable broadly to natural populations of *Drosophila* and other insect species? The use of fly strains from the stock center reared on a common cornmeal artificial medium and the same abiotic factors rather than wild-caught flies allowed us to standardize this large-scale experiment. Previous studies have shown that diet (*Fedina et al., 2012*), temperature (*Rajpurohit et al., 2021*), and mating status (*Everaerts et al., 2010*) can affect CHC profiles in *D. melanogaster*, therefore the CHC profiles that we obtained in our studies may not be representative of the CHC variations present in some of these species in their natural habitats. However, these differences in CHC profiles are mostly quantitative, that is, different quantities of the same CHCs, rather than different CHCs. Similarly, CHC variations between different natural populations of the same *Drosophila* species are also largely quantitative than qualitative (*Dembeck et al., 2015*; *Rusuwa et al., 2022*). Therefore, stock center strains would be representative of the qualitative CHC variations between species and our conclusions we reach in this study may be extended to wild populations. The use of standardized conditions also precludes the selection of species that require specialized media and conditions (e.g., species from the Hawaiian radiation and mycophagus species). While these are some of the limitations of our study, the standardized conditions allow us to reduce the number of confounds in our analyses for the correlation between CHC variation and desiccation resistance. As our study focused on a broad sampling of *Drosophila* and related species from different habitats spanning millions of years of evolution, the overall interpretation of the data and the broad conclusion that we reached regarding the correlation between the evolution of mbCHCs and desiccation resistance is likely to extend to natural populations of the *Drosophila* genus.

Constraints in the evolution of CHCs due to their roles in many insect species as contact pheromones are likely to affect their evolution in response to desiccation resistance. In this study, we showed that the evolution of mbCHCs are major determinants of desiccation resistance across *Drosophila* species, as *n*-alkanes shared the same branch in the fatty acyl-CoA pathway as monoenes and dienes (*Figure 5—figure supplement 8*), many of which function as contact pheromones in *Drosophila* (*Ferveur, 2005*). In other insect orders such as Lepidoptera, where volatile non-CHC pheromones are predominantly used for chemical communication (*Ando et al., 2004*), constraints on each class of CHCs may be different from *Drosophila*. Likewise, across eusocial hymenopterans (*Van Oystaeyen et al., 2014*), saturated CHCs such as *n*-alkanes and mbCHCs are used as queen pheromones and are likely to be under constraints in the evolution of desiccation resistance in these species. Finally, in some rare examples such as dragonflies where very-long-chain methyl ketones and aldehydes are the major cuticular wax components (*Futahashi et al., 2019*), the evolution of CHCs may not be a major factor in the evolution of desiccation resistance. While we limit our conclusions that the synthesis of mbCHCs with longer carbon-chain lengths is a common mechanism underlying the evolution of higher desiccation resistance to *Drosophila*, the evolution of cuticular components that modulate the melting temperature of the cuticular lipid layer can reduce water loss and increase desiccation resistance. We suggest that the evolution of CHC components in insects can be largely predictive of desiccation resistance between closely related species.

## Materials and methods
### *Drosophila* species and strains

In this study, 46 *Drosophila* species, as well as three *Scaptodrosophila* species and one *Chymomyza* species were either obtained from the National *Drosophila* Species Stock Center (NDSSC) or were gifts from various colleagues. Details are listed in *Supplementary file 1d*. The *D. melanogaster y w; attP40* strain and UAS-*Cyp4g1* RNAi strain were obtained from the Perrimon lab and the Bloomington *Drosophila* Stock Center, respectively. The 5'*mFAS*-GAL4 strain was constructed by cloning the oenocyte enhancer fragment of the *mFAS*/*FASN2* (*CG3524*) gene into the GAL4 vector *pBPGUw*

(Addgene #17575) using primers 5′CG3524-TOPO-F (5′-CACCCCGCGGCGTGTTATTGAACC-3′) and 5′CG3524-TOPO-R (5′-CTTGTTGCGCAGACAGACTG-3′), before injecting into the *D. melanogaster* strain 51C and integrated into the genome using the *PhiC31* integrase system. All species were reared on standard cornmeal medium (Flystuff 66-121 Nutri-Fly Bloomington Formulation). The phylogenetic relationship of all 50 species in this study was adapted from *Finet et al., 2021*. CHC- *D. melanogaster* were generated by crossing the 5′*mFAS*-GAL4 driver line (which expresses GAL4 in adult oenocytes) with the UAS-*Cyp4g1* RNAi strain (*Figure 5—figure supplement 9*).

## Experimental design

To investigate the contribution of CHCs to desiccation resistance, a cohort-based design was used. For each species, five to six cohorts were established and three measurements were conducted for each sex of the F1 progeny, including desiccation resistance, CHCs, and body weight (*Figure 1*). Each cohort was treated as a biological replicate. Each cohort in each species was established by pooling five females and five males on a standard cornmeal medium in the environmental chambers set at 25°C and a 12L:12D photoperiod. To maximize the food and spatial availability and minimize competition between the F1 progenies (*Mueller, 1988*), the parent flies in each cohort were transferred to fresh food after 5 days. F1 progeny flies were collected daily following emergence, separated by sex, and maintained on fresh cornmeal medium. All flies used for measurements were 4- to 5-day old.

## Desiccation resistance assays

Desiccation resistance assays were performed in a randomized and blinded manner with the setup consistent with a previously published protocol (*Figure 1*; *Chung et al., 2014*). Briefly, in each cohort, 10 adults of the same sex were subjected to the assay setup containing 10 g of silica gel (S7500-1KG, Sigma-Aldrich, St. Louis, MO). After the assays were assembled, all the setups were randomly arranged with a number assigned. Mortality of the flies was recorded hourly after 2 hr. For each cohort, one vial was scored and the average time in hours until all flies died was recorded as desiccation resistance. Desiccation resistance assays for the 50 *Drosophila* and related species were conducted at 25°C and with a 12L:12D photoperiod, while the desiccation assays for the CHC coating experiments were conducted at 27°C (GAL4/UAS) and a 12L:12D photoperiod.

## CHC analyses

CHCs were extracted and analyzed using GC–MS following previously published protocols (*Lamb et al., 2020*; *Savage et al., 2021*). Five flies of the same sex (4- to 5-day old) from each cohort were soaked for 10 min in 200 µl hexane containing hexacosane (C26; 25 ng/µl) as an internal standard. Extracts were directly analyzed by GC–MS (7890A, Agilent Technologies Inc, Santa Clara, CA) using a DB-17ht column (Agilent Technologies Inc, Santa Clara, CA). To identify the CHC composition, we first compared retention times and mass spectra to an authentic standard mixture (C7–C40) (Supelco 49,452U, Sigma-Aldrich, St. Louis, MO) with CHC samples. The types of CHCs, including methyl-branched alkanes, monoenes, dienes, and trienes, were then identified by a combination of their specific fragment ions on the side of functional groups (methyl branch or double bonds), retention times relative to linear hydrocarbon standards, and the *m/z* value of the molecular ion. The position of methyl branch in mbCHCs was determined using the protocol described in *Carlson et al., 1998*, while the position of double bonds was not determined in this study. Each CHC peak was quantified using its comparison with the peak area of the internal standard (C26) and represented as nanogram per fly (ng/fly). Because we observed a biased integration of peak areas on longer CHCs in running the standard mixture (*Figure 1—figure supplement 2*), we corrected the peak areas of CHCs based on the carbon-chain lengths using the integration from the standard mixture.

## Body weight measurement

Body weight was incorporated in the data collection and analysis. The body weight was determined as the difference between the Eppendorf tube containing 5–10 flies of the same sex and the same empty Eppendorf tube.

## Coating of synthetic compounds

To coat each of synthetic mbCHCs and *n*-alkanes on *D. melanogaster*, including 2MeC26, 2MeC28, 2MeC30, C23, C25, C27, C29, and C31, we first added 200 µl hexane containing 300 ng/µl of each CHC in a 2-ml glass vial and then used a nitrogen evaporator (BT1603 G-Biosciences) to evaporate the hexane with only mbCHCs precipitate at the bottom of the vial. For the control group, 200 µl hexane was added. After the hexane was evaporated, 10 flies of the same sex were transferred into each vial, following shaking on a Vortex for 20 s on, 20 s off, and 20 s on, following previous protocols (*Billeter et al., 2009*; *Chung et al., 2014*). The flies were then directly subjected to desiccation assays. Synthetic mbCHCs were kindly provided by Dr. Jocelyn Millar (University of California, Riverside) and synthetic *n*-alkanes were ordered from Sigma-Aldrich (C23 #91447-1G, C25 #286931-1G, C27 #51559-250MG, C29 #74156-250MG, and C31 #51529-250MG).

## Statistics

All analyses were conducted in R (Version 4.1). Correlation analyses were conducted with Pearson's method using '*cor.test*' function. The dependent variables were log-transformed to better conform with assumptions of normality. Variance in CHC beta diversity across desiccation resistance was determined using PERMANOVA using the 'vegan' package (*Anderson et al., 2006*). The random forest regression analysis was performed using the 'ranger' and 'randomForest' packages (*Liaw and Wiener, 2002*; *Wright and Ziegler, 2015*). The random forest regression models were built using both Out Of Bag estimate and test/training sets (70:30 split). To determine how useful each CHC variable is in the prediction of desiccation resistance in the random forest regression analysis, the importance of top predictor CHCs was quantified using permutation importance (*Altmann et al., 2010*). Desiccation resistance in flies coated with different mbCHCs was determined using one-way ANOVA at alpha = 0.05. Post hoc comparison was further conducted using Tukey's method. The ancestral trait reconstruction and estimation of phylogenetic signal, Pagel's $\lambda$, for mbCHC composition and desiccation resistance were determined using 'Phytools', 'Picante', and 'Rphylopars' packages (*Goolsby et al., 2017*; *Kembel et al., 2010*; *Revell, 2012*). The PGLS analysis was conducted using generalized least squares fit by maximum likelihood and the covariance structure between species was used under a Brownian motion process of evolution. The PGLS analyses were conducted using '*GLS*' function in '*ape*' package (*Paradis and Schliep, 2019*).

## Acknowledgements

We thank Dr. Jocelyn Millar for guidance and advice, as well as the synthetic compounds used in this study. We also thank Ye Ma, Elaina Giannetti, Taylor Hori, and Zhuo Chen for technical assistance, Yuzhang Shan for assistance with figure visualization, and the National *Drosophila* Species Stock Center for fly stocks of these different *Drosophila* species. This work is supported by a National Science Foundation grant (2054773) to H Chung.

## Additional information

### Funding

| Funder | Grant reference number | Author |
| --- | --- | --- |
| National Science Foundation | 2054773 | Henry Chung |

The funders had no role in study design, data collection, and interpretation, or the decision to submit the work for publication.

### Author contributions

Zinan Wang, Conceptualization, Data curation, Formal analysis, Validation, Investigation, Visualization, Methodology, Writing - original draft, Writing - review and editing; Joseph P Receveur, Software, Formal analysis, Validation, Methodology, Writing - review and editing; Jian Pu, Data curation, Validation, Investigation; Haosu Cong, Cole Richards, Data curation, Investigation; Muxuan Liang, Formal

analysis, Methodology, Writing - review and editing; Henry Chung, Supervision, Funding acquisition, Investigation, Methodology, Writing - original draft, Project administration, Writing - review and editing

### Author ORCIDs

Zinan Wang http://orcid.org/0000-0002-0509-4902
Henry Chung http://orcid.org/0000-0001-5056-2755

### Decision letter and Author response

Decision letter https://doi.org/10.7554/eLife.80859.sa1
Author response https://doi.org/10.7554/eLife.80859.sa2

## Additional files

### Supplementary files

• Supplementary file 1. Statistical output from phylogenetic comparative analyses and the list of species used in this study. (a) Phylogenetic signals for mbCHCs in *Drosophila* species. (b) Phylogenetic signals for desiccation resistance in *Drosophila* species. (c) Summary of the Phylogenetic Generalized Linear Square (PGLS) models between the longest mbCHCs and desiccation resistance for females and males in 50 *Drosophila* and related species. (d) List of species used in this study.

• MDAR checklist

• Source data 1. Data of normalized cuticular hydrocarbon (CHC) quantities, body weight, averaged desiccation resistance, total CHC quantity, and percentage of CHC quantities per body weight.

• Source code 1. Code for Pearson's correlation analysis and analysis of variance (ANOVA) tests in *Figure 3* and *Figure 5*.

• Source code 2. Code for NMDS plot and random forest analysis for *Figure 4*.

• Source code 3. Code for Phylogenetic Generalized Linear Square (PGLS) analyses in *Figure 6*.

### Data availability

All data generated or analyzed during this study are included in the manuscript and supporting source data file. Code used is uploaded as source codes 1-3.

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
