## [Editor Report]

These studies have presented convincing evidence that dessication resistance in *Drosophila* species is conferred primarily by long, methyl-branched cuticular hydrocarbons. These fundamental findings add significantly to our understanding of how *Drosophila* species have evolved to adapt and survive in different environments. This study will be of interest to insect physiologists and ecologists as well as evolutionary biologists.

---

## [Decision Letter]

**Decision letter after peer review:**

Thank you for submitting your article "Desiccation resistance differences in *Drosophila* species can be largely explained by variations in cuticular hydrocarbons" for consideration by *eLife*. Your article has been reviewed by 3 peer reviewers, and the evaluation has been overseen by a Reviewing Editor and K VijayRaghavan as the Senior Editor. The reviewers have opted to remain anonymous.

Essential revisions:

1) Perform control coating experiments and "replacement" experiments as outlined by Reviewer 1 in order to validate your hypothesis that the longer methyl-branched CHCs are critical to desiccation resistance.

2) Discuss the differences seen between males and females and the role that the methyl-branched cuticular pheromones may be playing.

3) Discuss the implications of using lab-reared flies for this analysis rather than wild-caught flies for the overall interpretation of the data.

4) Add a short discussion of how this study on *Drosophila* species may relate to desiccation resistance in other insects – mention similarities and differences.

*Reviewer #1 (Recommendations for the authors):*

A central issue in this work is to lift the correlative nature of the impressive data to the level of causativity. The coating experiment performed is, to me, insufficient to do so. I propose "relacement" and control experiments to support this point.

Despite this, there is another crucial point that needs to be addressed more convincingly: there are quite a few pieces of evidence published that mbCHC may be a major type of CHCs that is involved in desiccation resistance. So, the findings here are not really new. Admittedly, the amount of data/species is impressive; nevertheless, the statstical data (relatively low r-values) are not doubtlessly clarifying this matter. Indeed, this is a complex problem (trade-off of the CHC composition), and there won't be any simple solution to it. I will happy to discuss this with the authors.

*Reviewer #2 (Recommendations for the authors):*

This was a well-organized manuscript, and the authors do not go beyond their data in reaching their conclusions.

*Reviewer #3 (Recommendations for the authors):*

I have several suggestions for the authors as indicated below about the general scope of the study and interpretation of how this study could relate to other insects. The way this paper is written, it is most relevant to studies on *Drosophila* and may not pertain to other insects.

An aspect that is not discussed is the fact that many insects have internally branched methyl groups. It is surprising to me that these *Drosophila* species only have 2 methyl branched hydrocarbons. Although 2meCHC does have the highest melting temperature. Internally branched methyl groups have lower melting temperatures and would thus not contribute as much to desiccation. The discussion could include the contribution of other lipids to desiccation rates in other insects.

Line 66 – The authors describe the general mechanism in conserving water and preventing desiccation as a lipid layer found on the epicuticle and they named it cuticular hydrocarbons. This is not accurate because there are other types of lipids that could be found on the surface in addition to hydrocarbons. In some insects the majority of the lipids are hydrocarbons but other insects have few hydrocarbons but other types of lipids. The term cuticular lipids refers to a broader category of lipids and not just to hydrocarbons.

Line 121 – In addition to other types of lipids that could be found on the cuticular surface not all insects would have methyl branched hydrocarbons. Therefore this might not be a general mechanism to evolve higher levels of desiccation resistance across Insecta.

Line 258 – Compare *D. immigrans* vs. *D. mojavensis*. Both apparently have similar quantities of 2meC32 yet *D. immigrans* is not very desiccation resistant. Indicating that other factors are involved and is not due to just 2meC32.

Line 375 – Were body weight measurements take on dead or alive flies?

Materials and methods – Statistics

I cannot comment on the statistics as I am not familiar with most of the statistical packages used in the study.

---

## [Author Response]

Essential revisions:1) Perform control coating experiments and "replacement" experiments as outlined by Reviewer 1 in order to validate your hypothesis that the longer methyl-branched CHCs are critical to desiccation resistance.

Reviewer 1 has suggested two different sets of experiments to be performed on flies without CHCs (CHC- flies). First, “control” experiments which coats “CHCs with the same chain lengths but without branched methyl groups” i.e., n-alkanes of different lengths on CHC- flies. Second, “replacement” experiments which coats mbCHCs of different lengths on CHC- flies. To address the reviewer’s comment, we performed the “control” experiments on both WT and CHC- flies and the “replacement” experiments on the CHC- flies. In summary, we performed the following experiments and these new data are added in a new Figure 5—figure supplement 2:

Initial submission

[WT +mbCHCs]

Resubmission

[WT +mbCHCs from initial submission], [WT +n-alkanes], [CHC- +mbCHCs], [CHC- +n-alkanes]

a) In our original submission, we showed that the coating of longer mbCHCs, 2MeC28 and 2MeC30, on wildtype (WT) *D. melanogaster* flies can increase desiccation resistance, while the shorter 2MeC26 does not have a significant effect (Figure 5E). We now provide additional analyses testing the correlation between mbCHC length and desiccation resistance and show that the significant increase in desiccation resistance is positively correlated with longer mbCHCs (Figure 5—figure supplement 2), suggesting that longer mbCHCs are able to confer higher desiccation resistance in WT flies. We obtained similar results with the “control” experiment, showing that the coating WT *D. melanogaster* flies with alkanes (C23, C25, C27, C29, C31) is able to increase desiccation resistance which is positively correlated with carbon chain length (Figure 5—figure supplement 2, Figure 5—figure supplement 3). This is not surprising as alkanes are saturated CHCs with high melting temperatures.

b) In the new “replacement” experiments, we showed that the coating of each of the mbCHCs, 2MeC26, 2MeC28, and 2MeC30, is able to significantly increase desiccation resistance in CHC- flies of both sexes (Figure 5—figure supplement 5). We also showed that the coating of all of the alkanes (C23, C25, C27, C29, C31) is able to significantly increase desiccation resistance in female CHC- flies, but only C31 is able to significantly increase desiccation resistance in male CHC- flies (Figure 5—figure supplement 5). However, desiccation resistance of these CHC- flies are very low and the coating of these synthetic CHCs did not increase desiccation resistance by much, albeit statistically significantly. Only a weak positive correlation between longer mbCHCs and higher desiccation was observed in female CHC- flies. There is no correlation between mbCHC length and higher desiccation in male CHC- flies and no correlation between alkane length and higher desiccation in female and male CHC- flies (Figure 5—figure supplement 6). This is similar with Krupp et al. 2020 where the authors showed coating CHC- with either only alkanes or mbCHCs only produce small increases in desiccation resistance, while coating CHC- flies with full WT extracts is able to result in a much larger increase in desiccation resistance. Both Krupp et al. and our observations are consistent with previous hypotheses that the CHC layer work as a blend rather than as individual components (Gibbs, 1998; Menzel et al., 2019; Wigglesworth, 1945).

Although we showed longer chain *n-*alkanes are also positively correlated with higher desiccation resistance when perfumed on WT flies, these CHCs are not widely found in *Drosophila* beyond the *melanogaster* group, thus were not ranked highly in our random forest analysis. In other words, we again suggest that longer chain *n-*alkanes are able to confer higher desiccation resistance, but do not contribute to the evolution of higher desiccation resistance in *Drosophila*. Therefore, we reiterate the conclusion that longer mbCHCs are critical for desiccation resistance across *Drosophila* but will also add the caveat that mbCHCs alone (in the absence of all the other CHCs in the profile of a *Drosophila* fly) do not confer high desiccation resistance.

We have modified added new results manuscript (Lines 222 to 250). In addition, we have modified the Materials and methods to include the new experiments (Lines 410 to 419) that uses a slightly different but highly effective method in generating CHC- flies.

References

Gibbs, A.G. (1998). Water-proofing properties of cuticular lipids. American Zoologist *38*, 471-482.

Krupp, J.J., Nayal, K., Wong, A., Millar, J.G., and Levine, J.D. (2020). Desiccation resistance is an adaptive life-history trait dependent upon cuticular hydrocarbons, and influenced by mating status and temperature in *D. melanogaster*. Journal of Insect Physiology *121*, 103990.

Menzel, F., Morsbach, S., Martens, J.H., Räder, P., Hadjaje, S., Poizat, M., and Abou, B. (2019). Communication versus waterproofing: the physics of insect cuticular hydrocarbons. Journal of Experimental Biology *222*.

Wigglesworth, V.B. (1945). Transpiration through the cuticle of insects. Journal of Experimental Biology *21*, 97-114.

2) Discuss the differences seen between males and females and the role that the methyl-branched cuticular pheromones may be playing.

To address this comment, we have split the comment into two parts: (a) discuss the differences seen between males and females and (b) discuss the role that the methyl-branched cuticular pheromones may be playing.

(a) We have added the following section to the Discussion section (Lines 297 to 310) of our manuscript to discuss the differences seen between males and females:

“Our overall analyses support the conclusion that the evolution of longer mbCHCs is correlated with higher desiccation resistance in our dataset, but we observed minor differences between males and females (Figures 3 and 5). In particular, while higher quantities of 2MeC28 and 2MeC32 are both positively correlated with higher desiccation resistance in both males and females (Figure 5B, D), there are differences between the sexes regarding 2MeC26 and 2MeC30 (Figure 5A, C). There could be several underlying reasons for this. Firstly, across our dataset, CHCs are sexually dimorphic in many species, leading to overall differences in the correlation analyses between the sexes. Secondly, other physiological differences, such as size differences, may contribute to the differences between males and females. In many species such as *D. melanogaster*, females are larger than males and this may contribute to the higher basal desiccation resistance (when CHCs are taken out of the equation) of female CHC- flies (3.6 ± 0.1 h) compared to that of male CHC- flies (2.1 ± 0.1 h) (Figure 5E). While these are minor differences in the overall correlation analyses between males and females, the synthetic CHC coating experiments showed that coating of longer mbCHCs confers higher desiccation resistance in both males and females.”

(b) We have expanded the discussion regarding sexual antagonism in *D. serrata* to discuss the roles that methyl-branched cuticular pheromones could be playing in affecting the differences in the evolution of mbCHCs and desiccation resistance (Lines 359 to 367):

“In *D. serrata*, having higher amounts of 2MeC26 leads to higher mating success in male *D. serrata* (Chung et al., 2014). *Northern* populations of this species produce lower amounts of 2MeC26 but higher amounts 2MeC30, due to a polymorphism in a fatty acyl-CoA reductase gene that shifts to an overall production of longer CHCs. As CHCs are sexually monomorphic in *D. serrata*, this polymorphism led to lower mating success in male *northern* flies and higher desiccation resistance in female *northern* flies compared to their common counterparts (Rusuwa et al., 2022). The sexually antagonistic effect due to the functions of the different mbCHCs may affect their evolutionary response to desiccation in many species.”

References

Chung, H., Loehlin, D.W., Dufour, H.D., Vaccarro, K., Millar, J.G., and Carroll, S.B. (2014). A single gene affects both ecological divergence and mate choice in *Drosophila*. Science *343*, 1148-1151.

Rusuwa, B.B., Chung, H., Allen, S.L., Frentiu, F.D., and Chenoweth, S.F. (2022). Natural variation at a single gene generates sexual antagonism across fitness components in *Drosophila*. Current Biology *32*, 3161-3169.

(3) Discuss the implications of using lab-reared flies for this analysis rather than wild-caught flies for the overall interpretation of the data.

We agree with the reviewer’s comment that the CHC composition found in lab-reared flies may not be reflective of CHC variation of individual species in the field. In published studies, almost all the CHC variation between different populations of a single species are quantitative differences, rather than qualitative differences in CHCs. In addition, as our study focused on a broad sampling of *Drosophila* and related species from different habitats spanning millions of years of evolution, rather than individual species, the overall interpretation of the data and the broad conclusion that we reached regarding the correlation between the evolution of mbCHCs and desiccation resistance is likely to extend to natural populations or wild-caught flies of the *Drosophila* genus. We have acknowledged the reviewer’s comment in the paper and added discussion points regarding this in the text (Lines 369 to 387):

“How much of what we learnt in this study is applicable broadly to natural populations of *Drosophila* and other insect species? The use of fly strains from the stock center reared on a common cornmeal artificial medium and the same abiotic factors rather than wild-caught flies allowed us to standardize this large-scale experiment. Previous studies have shown that diet (Fedina et al., 2012), temperature (Rajpurohit et al., 2021), and mating status (Everaerts et al., 2010) can affect CHC profiles in *D. melanogaster*, therefore the CHC profiles that we obtained in our studies may not be representative of the CHC variations present in some of these species in their natural habitats. However, these differences in CHC profiles are mostly quantitative, i.e., different quantities of the same CHCs, rather than different CHCs. Similarly, CHC variations between different natural populations of the same *Drosophila* species are also largely quantitative than qualitative (Dembeck et al., 2015; Rusuwa *et al.*, 2022). Therefore, stock center strains would be representative of the qualitative CHC variations between species and our conclusions we reach in this study may be extended to wild populations. The use of standardized conditions also preclude the selection of species that require specialized media and conditions (e.g., species from the Hawaiian radiation and mycophagus species). While these are some of the limitations of our study, the standardized conditions allow us to reduce the number of confounds in our analyses for the correlation between CHC variation and desiccation resistance. As our study focused on a broad sampling of *Drosophila* and related species from different habitats spanning millions of years of evolution, the overall interpretation of the data and the broad conclusion that we reached regarding the correlation between the evolution of mbCHCs and desiccation resistance is likely to extend to natural populations of the *Drosophila* genus.”

(4) Add a short discussion of how this study on *Drosophila* species may relate to desiccation resistance in other insects – mention similarities and differences.

We have added a new section (Line 388 to 404) at the end of the discussion to discuss how our study may relate to desiccation resistance in other insects including some differences and similarities:

“Constraints in the evolution of CHCs due to their roles in many insect species as contact pheromones are likely to affect their evolution in response to desiccation resistance. In this study, we showed that the evolution of mbCHCs are major determinants of desiccation resistance across *Drosophila* species, as *n*-alkanes shared the same branch in the fatty acyl-CoA pathway as monoenes and dienes (Figure S8), many of which function as contact pheromones in *Drosophila* (Ferveur, 2005). In other insect orders such as Lepidoptera, where volatile non-CHC pheromones are predominantly used for chemical communication (Ando et al., 2004), constraints on each class of CHCs may be different from *Drosophila*. Likewise, across eusocial hymenopterans (Van Oystaeyen *et al.*, 2014), saturated CHCs such as *n*-alkanes and mbCHCs are used as queen pheromones and are likely to be under constraints in the evolution of desiccation resistance in these species. Finally, in some rare examples such as dragonflies where very long-chain methyl ketones and aldehydes are the major cuticular wax components (Futahashi et al., 2019), the evolution of CHCs may not be a major factor in the evolution of desiccation resistance. While we limit our conclusions that the synthesis of mbCHCs with longer carbon-chain lengths is a common mechanism underlying the evolution of higher desiccation resistance to *Drosophila*, the evolution of cuticular components that modulate the melting temperature of the cuticular lipid layer can reduce water loss and increase desiccation resistance. We suggest that the evolution of CHC components in insects can be largely predictive of desiccation resistance between closely related species.”

Reviewer #1 (Recommendations for the authors):A central issue in this work is to lift the correlative nature of the impressive data to the level of causativity. The coating experiment performed is, to me, insufficient to do so. I propose "relacement" and control experiments to support this point.Despite this, there is another crucial point that needs to be addressed more convincingly: there are quite a few pieces of evidence published that mbCHC may be a major type of CHCs that is involved in desiccation resistance. So, the findings here are not really new. Admittedly, the amount of data/species is impressive; nevertheless, the statstical data (relatively low r-values) are not doubtlessly clarifying this matter. Indeed, this is a complex problem (trade-off of the CHC composition), and there won't be any simple solution to it. I will happy to discuss this with the authors.

For many of the data in the Figures 3 and 5, we use the Pearson correlation test to identify correlations between our variables. The validity of the Pearson correlation test does not rely on the linear assumptions. When the relationship is close to be linear, the Pearson correlation test will have typically higher power than other nonparametric tests, i.e., it is capable of detecting possible monotone relationship close to a linear relationship with fewer samples. However, we agree with the reviewer that the slopes might look misleading. Therefore, we have removed the lines/slopes from the figures.

Reviewer #3 (Recommendations for the authors):I have several suggestions for the authors as indicated below about the general scope of the study and interpretation of how this study could relate to other insects. The way this paper is written, it is most relevant to studies on *Drosophila* and may not pertain to other insects.An aspect that is not discussed is the fact that many insects have internally branched methyl groups. It is surprising to me that these *Drosophila* species only have 2 methyl branched hydrocarbons. Although 2meCHC does have the highest melting temperature. Internally branched methyl groups have lower melting temperatures and would thus not contribute as much to desiccation. The discussion could include the contribution of other lipids to desiccation rates in other insects.Line 66 – The authors describe the general mechanism in conserving water and preventing desiccation as a lipid layer found on the epicuticle and they named it cuticular hydrocarbons. This is not accurate because there are other types of lipids that could be found on the surface in addition to hydrocarbons. In some insects the majority of the lipids are hydrocarbons but other insects have few hydrocarbons but other types of lipids. The term cuticular lipids refers to a broader category of lipids and not just to hydrocarbons.Line 121 – In addition to other types of lipids that could be found on the cuticular surface not all insects would have methyl branched hydrocarbons. Therefore this might not be a general mechanism to evolve higher levels of desiccation resistance across Insecta.Line 258 – Compare *D. immigrans* vs. *D. mojavensis*. Both apparently have similar quantities of 2meC32 yet *D. immigrans* is not very desiccation resistant. Indicating that other factors are involved and is not due to just 2meC32.

The reviewer is correct. While our model showed that CHC composition was able to explain 85.5% of the variation in desiccation resistance, it does not attribute all the variation in desiccation resistance to CHC composition. We agree suggest there are other physiological factors that also can contribute to desiccation resistance across the *Drosophila* genus.

Line 375 – Were body weight measurements take on dead or alive flies?

The body weight measurements were taken on alive flies.